# The Possible Roles of Glucosamine-6-Phosphate Deaminases in Ammonium Metabolism in Cancer

**DOI:** 10.3390/ijms252212054

**Published:** 2024-11-09

**Authors:** Roberto Lara-Lemus, Manuel Castillejos-López, Arnoldo Aquino-Gálvez

**Affiliations:** 1Departamento de Biomedicina Molecular e Investigación Traslacional, Instituto Nacional de Enfermedades Respiratorias Ismael Cosío Villegas (INER), Mexico City 14080, Mexico; 2Departamento de Bioquímica, Facultad de Medicina, Universidad Nacional Autónoma de México (UNAM), Mexico City 04510, Mexico; araquiga@yahoo.com.mx; 3Departamento de Epidemiología e Infectología Hospitalaria, Instituto Nacional de Enfermedades, Respiratorias Ismael Cosío Villegas (INER), Mexico City 14080, Mexico; mcastillejos@gmail.com; 4Laboratorio de Biología Molecular, Departamento de Fibrosis Pulmonar, Instituto Nacional de Enfermedades Respiratorias Ismael Cosío Villegas (INER), Mexico City 14080, Mexico

**Keywords:** glucosamine-6-phosphate deaminases, ammonium, cancer

## Abstract

Nearly 5% of the glucose-6-phosphate (Glc6P) in cells is diverted into the hexosamine biosynthetic pathway (HBP) to synthesize glucosamine-6-phosphate (GlcN6P) and uridine diphosphate *N*-acetyl-glucosamine-6-phosphate (UDP-GlcN6P). Fructose-6-phosphate (Fru6P) is a common intermediary between glycolysis and the HBP. Changes in HBP regulation cause abnormal protein N-glycosylation and *O*-linked-N-acetylglucosamine modification (O-GlcNAcylation), affecting protein function and modifying cellular responses to signals. The HBP enzymes glucosamine-6-phosphate deaminases 1 and 2 (GNPDA1 and 2) turn GlcN6P back into Fru6P and ammonium, and have been implicated in cancer and metabolic diseases. Despite the plentiful literature on this topic, the mechanisms involved are just beginning to be studied. In this review, we summarize, for the first time, the current knowledge regarding the possible roles of the isoenzymes of both GNPDAs in the pathogenesis and development of metabolic diseases and cancer from a molecular point of view, highlighting their importance not only in supplying carbon from glycolysis, but also in ammonia metabolism.

## 1. Introduction

Hexosamine metabolism has been studied for a long time in mammals, protozoans, bacteria, and fungi. However, in the past two decades, it has gained more relevance due to significant advances in our recognition of the role of the post-translational modification of *O*-linked β-*N*-acetylglucosamine (*O*-GlcNAc) in proteins. *O*-GlcNAc proteins have regulatory functions in several processes, such as metabolism, cell cycle control, histone methylation, and oncogenesis [1]. *N*-Acetyl-D-glucosamine (GlcNAc) is bound to the hydroxyl group of serine or threonine residues in target proteins from uridine diphosphate *N*-acetylglucosamine (UDP-GlcNAc). Intermediaries from other metabolic routes, such as Fru6P, glutamine (Gln), acetyl-Coenzyme A (AcCoA), and uridine-5′-triphosphate (UTP) [2], participate in the synthesis of this molecule (Figure 1). Thus, the HBP is connected not only to glycolysis, but also lipids, amino acids, and nucleotide metabolism [3]. In this sense, a better knowledge of the enzymes and regulatory mechanisms of the HBP is essential for improving our understanding of cancer and metabolic diseases. In this review, we focus on the allosteric enzyme glucosamine-6-phosphate isomerase deaminase (GNPDA, EC. 3.5.99.6), which catalyzes the reversible conversion of GlcN6P into Fru6P and ammonium (NH_4_^+^) [4], as follows: Fru6P + NH_4_^+^ ↔ GlcN6P + H_2_O

Two different genes encoding the enzymes GNPDA1 and 2 have been identified in mammal genomes [5,6]. Recent studies have implied the involvement of these GNPDAs in some human diseases. As GNPDAs localize in metabolic intersections with glycolysis, in cancer cells, GNPDAs play a key role as suppliers of carbon for glycolysis [3]. In the same way, GNPDA2 can provide glycolytic precursors for tri-acyl glyceride synthesis, with this mechanism having been suggested as a possible explanation for the relationship between GNPDA2 and obesity. This review focuses on the survival of cancer cells in environments with high concentrations of NH_4_^+^, as the other product of GNPDA reactions.

**Figure 1 ijms-25-12054-f001:**
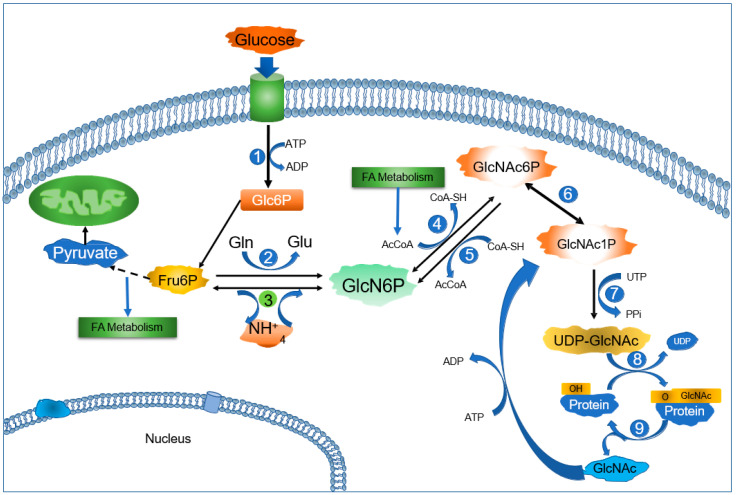
Graphic description of the HBP showing the position of GNPDAs. Glucose, a key player in the metabolic pathways, is transported into the cell and then phosphorylated by hexokinase (1). It is then isomerized into Fru6P and follows the glycolytic pathway until pyruvate. GlcN6P, a significant metabolite, is synthesized from Fru6P and glutamine (Gln) by GFAT (Glutamine-fructose-6-phosphate transaminase or glucosamine-6-phosphate synthase) (2) or from NH_4_^+^ instead of Gln by the reverse reaction of GNPDA (3). GNPDA activity is a crucial part of the HBP, and the catabolic direction of this reaction is favored. GlcN6P is acetylated from Acetyl-CoA to form GlcNAc6P, the allosteric activator of GNPDA, by the enzyme Glucosamine-Phosphate N-Acetyltransferase 1 (4). Deacetylation can occur according to a reverse reaction or be driven by the enzyme N-acetylglucosamine-6-phosphate deacetylase (5). Phosphoacetylglucosamine mutase isomerizes GlcNAc6P into GlcNAc1P (6). Next, the UDP-N-acetyl hexosamine pyrophosphorylase produces UDP-GlcNAc from UTP and GlcNAc-1P (7). O-GlcNAc transferase (OGT) transfers the O-GlcNAc moiety to specific serine/threonine residues of diverse proteins (8). Finally, the enzyme Protein-O-GlcNAc hydrolase (OGA) removes GlcNAc from the modified protein (9). GlcNAc can be phosphorylated from ATP by a kinase to regenerate GlcNAc-1P. The fate of this metabolite depends on several conditions, such as the requirements of hyaluronan synthesis [1,7]. It can be recycled back into UDP-GlcNAc or to GlcN6P; then, GNPDA (3) can use the carbon in glycolysis and the ammonium in other biosynthetic pathways.

## 2. Glucosamine-6-Phosphate Deaminases: A Short Story

In mammals, the activity of GNPDA was first described in 1956—in the kidneys by Leloir and Cardini [4] and in the brain by Faulkner and Quastel [8]. From then on, efforts were focused on the purification of this enzyme, aimed at determining its molecular, structural, and kinetic properties. Some publications reported partial purifications and kinetic studies with animal tissues taken from the human brain [9], pig and rat kidneys [10,11,12], and flies [13,14], as well as from human and rabbit erythrocytes [15,16]. At this point, no GNPDA genes had been cloned, and researchers faced considerable problems in obtaining pure and homogeneous enzymes from mammal sources. This led to misinterpretations of results, such as the band patterns obtained from SDS-denaturing acrylamide gel electrophoresis. The presence of more than a single band was described by Kikuchi et al., 1979, as demonstrating the instability of the purified enzyme [11] or that the enzyme was composed of two kinds of highly homologous sub-units (α and β) with slightly different molecular weights, as established by Cayli et al. [17]. These results contrasted with those obtained from bacteria, particularly *Escherichia coli* (*E. coli*), which have a single GNPDA gene (NagB), and the number of purification, crystallographic, and metabolic studies on *E. coli* grew rapidly within a few years [18,19,20,21,22,23]. By 1992, the first report was published showing the purification to homogeneity of the mammal GNPDA from a dog kidney cortex (dGNPDA); the purity of the protein was demonstrated by a single band both in non-denaturing and SDS-denaturing acrylamide gel electrophoresis, isoelectric focusing, and as a single peak in chromatofocusing chromatography [24].

Like bacterial deaminase, the dog kidney enzyme is also a homo-hexameric protein, but the mammal enzyme showed a slightly higher molecular weight of the monomer (30.5 vs. 29 kDa); subsequently, it was shown to be an isoform corresponding to GNPDA2. Soon after, the first mammal GNPDA gene was cloned. Notably, it came from a misinterpretation of some experimental results, which assigned the modulation of calcium oscillations in hamster eggs to GNPDA [25]. Afterward, some reports, including one from the authors of the hamster report, demonstrated that this enzyme lacked any oscillin (as it was named) activity. The molecular characterizations of the human, hamster, and mouse variants of the GNPDA gene were published in [25,26,27,28,29].

In summary, a single gene was cloned from different mammal sources, corresponding to the GNPDA1 enzyme. By then, molecular, kinetic, and allosteric analyses of an endogenous GNPDA (non-recombinant or modified enzyme) had been performed in GNPDA purified from a beef kidney (bGNPDA) [30]. This enzyme is also homo-hexameric, but with a higher molecular weight (32.5 kDa for the monomer) than that of dog deaminase. Notably, this enzyme displayed a very different kind of allosteric activation by GlcNAc6-P than that reported for the *E. coli* and dog kidney enzymes [17,20]. The amino-terminal sequence revealed that it corresponded to GNPDA1. It was difficult to explain how a single GNPDA gene could yield two different enzymes catalyzing the same reaction, but with different molecular and kinetic properties. This paradox was solved almost simultaneously by Zhang et al. and Arreola et al. when they reported the existence of a second GNPDA gene [5,6]. Considering this, the results reported by Cayli et al. [17] could be understood as two isoenzymes rather than one enzyme with two different sub-units. The research landscape of amino sugar metabolism, and particularly of the deaminase enzyme, suddenly changed now that there were two isoenzymes, GNPDA1 and GNPDA2, both highly homologous and homo-hexameric, catalyzing the same reaction but displaying different kinetic and molecular features. This allowed the possibility of there being distinctive functions for each isoenzyme in cell metabolism and physiology, with each being related differently to diverse human diseases.

## 3. GNPDAs Show Relevant Differences in Enzyme Kinetics

Significant differences in the kinetic behavior of the GNPDA isoenzymes are likely relevant for their physiologic and pathologic roles in mammal cells. A suitable starting point for this section is mentioning that GNPDA from *E. coli* (eGNPDA) is the most studied enzyme, both structurally and kinetically [18,19,20,21,22,23]. However, this enzyme has been studied in other bacteria and organisms [31,32,33,34,35]. The allosteric effector is GlcNAc6P, which activates eGNPDA by decreasing the Michaelis constant (*kM*) without modifying the maximum velocity (*Vmax*) as the activator concentration increases. Next, it is important to emphasize that when many of the publications cited in the previous section reported values for *kM* and *Vmax*, it is likely that those data were obtained from mammal tissues and cell lines containing both GNPDA isoenzymes; thus, the reported values are *kM* and *Vmax* (*kM*, *Vmax* app). On the other hand, kinetic assays were performed in the presence of the allosteric activator GlcNAc6P, which has an intense effect, increasing affinity for the substrate GlcN6P. These issues could explain why the reported *kM* values are quite similar [9,15,24,30,36]. As mentioned, mammal GNPDA1 was first characterized from a genic point of view, and kinetic studies specific to this purified isoenzyme are only available in two publications [30,36]. bGNPDA1 displays an uncommon kind of allosteric activation—in the absence of GlcNAc6P, the enzyme shows hyperbolic kinetics as the concentration of the substrate GlcN6P increases, but the catalytic constant (*kcat*) is tiny (*kcat* = 0.37 s^−1^). GlcNAc6P increases the *Vmax* of the enzyme and, at saturation, *kcat* reaches 10.1 s^−1^; that is, the allosteric transition produces a 27-fold increase in *Vmax* without affecting *kM*, which classifies bGNPDA1 as an allosteric enzyme of the *Vmax* or *V*-type [30]. In the reverse sense of the reaction, this enzyme also displays an intense *V*-effect, and it is practically inactive in the absence of the allosteric activator over a wide range of Fru6P and NH_4_^+^ concentrations [30]. The *kM* values reported in the presence of its allosteric activator were 5.9 and 3.7 mM for Fru6P and NH_4_^+^, respectively. The *kM* for NH_4_^+^ is lower than eGNPDA (31.4 mM), suggesting that this reverse reaction could play a biosynthetic role in bovine kidneys. A deeper kinetic study of GNPDA1 was performed in human recombinant enzyme (hrGNPDA1). In this case, GlcNAc6P affected both the *kM* and *Vmax* of the enzyme, behaving as a mixed allosteric enzyme (*K* and *V*) [36]; the authors named this behavior “antiergistic”, because GlcNAc6P has an inhibitory effect upon *kM*, but it activates the enzyme by increasing the *Vmax*. This is in contrast to “synergistic”, because, in this case, activation effects on both *kM* and *Vmax* can be expected [30]. In the absence of GlcNAc6P, both bGNPDA1 and hrGNPDA are almost inactive (showing a very low *kcat*), but small amounts of GlcNAc6P increase *Vmax* significantly, so the allosteric effector is described as an essential activator. In nature, allosteric *V*-enzymes are very scarce, and they are true mixed systems (*K*/*V*); so, it is possible that bGNPDA1 could also be an allosteric mixed system. 

From a kinetic point of view, GNPDA2 exhibits similarities to eGNPDA. dGNPDA2 is a typical allosteric *K*-enzyme—it is activated by its allosteric ligand, which increases the affinity for its substrate [24]. In this case, GlcNAc6P is not an essential activator, because GNPDA2 is active in the absence of GlcNAc6P. dGNPDA2 has a *kM* value for GlcN6P of about one order of magnitude lower than the enzyme from E. coli (0.25 and 2.0 mM, respectively), but its *kcat* is lower (250 s^−1^ and 1800 s^-l^, respectively) [18,24]. As a result, the catalytic efficiency (*kcat*/*kM*) for both enzymes is quite similar. No dGNPDA2 *kM*s have been reported for Fru6P or NH_4_^+^, leaving the possibility that these values are close to the bacterial enzyme; however, this remains to be experimentally revealed.

The distinct responses of GNPDAs to GlcNAc6P could have significant metabolic implications. The activity of GNPDA1 is almost entirely dependent on the level of GlcNAc6P in the cell (essential allosteric activator) [30,36]. When GlcNAc6P levels increase, either due to the enhanced synthesis or accelerated degradation of UDP-GlcNAc, GNPDA1 works in the deaminating sense, using the carbon atoms of GlcN6P in glycolysis, yielding ATP or feeding tri-acyl glyceride metabolism (Figure 2). This increase in GlcNAc6P levels also leads to an increase in the concentration of NH_4_^+^, which can be passed into the bloodstream or sent to biosynthetic pathways, as discussed later. Due to the low *kM*s for Fru6P and NH_4_^+^, the GNPDA1 reaction could be easily reversible and contribute to GlcNAc6P synthesis [6,13,14,30,34]. On the other hand, GNPDA2 is more dependent on GlcN6P concentrations than GlcNAc6P, suggesting that this isoenzyme, along with GFAT, fuels the HBP.

## 4. GNPDAs and Disease

GNPDAs are associated with several diseases, including obesity and metabolic malfunctions, neurologic diseases, and cancer, primarily gastrointestinal cancer (Table 1). From Table 1, it is evident that each GNPDA is related to different kinds of pathologies. The GNPDA2 gene is associated with a greater body mass index and obesity, while colorectal cancer, hepatocellular, and other gastrointestinal malignancies show increased GNPDA1 protein levels; the latter is addressed in the next section.

Nearly all the reports linking GNPDA2 with obesity are based on genome-wide association studies (GWASs) conducted in different populations worldwide (Table 1). These studies identified an obesity-related single-nucleotide polymorphism (SNP), rs10938397, near to the GNPDA2 gene, which is associated with an increased body mass index and obesity; in fact, the relationship of GNPDA2 with type 2 diabetes and asthma is mediated through obesity. Due to the nature of these studies (primarily meta-analyses), they do not provide insights into the possible mechanisms by which GNPDA2, as an enzyme or protein, could be involved in the pathology of obesity; however, the few experimental publications in this field provide valuable data. *GNPDA2* and four other genes (TMEM18, KCTD15, SH2B1, and NEGR1) are highly expressed in the hypothalamus [60]; it was recently shown in a mouse model that GNPDA2 in the hypothalamus does not control appetite, but regulates glucose homeostasis. The inhibition of GNPDA2 in the hypothalamus did not affect either food intake or body weight [61]. The only study on the metabolic role of GNPDA2 demonstrated that the over-expression of GNPDA2 enhances adipogenesis and lipid accumulation in adipose-derived mesenchymal stem cells [52]. The over-expression of this gene positively affects the mRNA level of peroxisome proliferator-activated receptor-γ (PPAR-γ) and signal transducer and activator of transcription 5 (STAT5), enhancing adipocyte differentiation [52] (Figure 3). Another notable finding from this study was the increased expression of genes related to non-alcoholic fatty liver disease (NAFLD) when the GNPDA2 gene was silenced.

In summary, GNPDA2 promotes lipogenesis and adipocyte proliferation, which is in accordance with the weight gain and obesity found in multiple GWAS and epidemiologic studies (Table 1); however, in the central nervous system, this enzyme does not increase food intake. It is important to recall that there is no direct evidence that the SNP rs10938397 alters endogenous *GNPDA2* expression in adipose tissue; however, changes in glucose homeostasis could be associated with a higher risk of type 2 diabetes. Overall, these results indicate the involvement of the GNPDA2 isoenzyme in the metabolic changes underlying type 2 diabetes and some other related pathologies such as NAFLD.

## 5. GNPDA1 and Cancer

Recently, evidence showing a clear relationship between GNPDA1 and some cancers was described (Table 1). However, the first indication that any GNPDA was related to cancer probably arose from a report by Sukeno et al. in 1971 [64]. In this report, they found an increased synthesis of glycogen and lactate in a rat model of Yoshida sarcoma cells. When incubated with 2 mM of glucosamine (GlcN), the cells showed an accumulation of GlcN6P and a rapid transformation into glycogen and lactate, with GNPDA being identified as the enzyme responsible. These cells had higher amounts of GNPDA than hepatic cells. From this, we can postulate that GNPDA1 is activated because higher concentrations of GlcN6P can be transformed in GlcNAc6P, producing more Fru6P and NH_4_^+^. Eight years later, Kikuchi et al. reported an increase of more than ten times in GNPDA enzymatic activity in rat hepatoma cells with respect to normal liver tissue. They also found a change in the fate of *N*-acetylglucosamine (GlcNAc)—while in normal hepatocytes, it is transformed mainly into sialic acid (biosynthetic route), hepatoma cells transform this amino sugar into glycolytic intermediates (catabolic pathway) [65]. Once again, in transformed cells, GNPDA1 activity can help in obtaining energy through glycolysis. Forty-one years later, it was confirmed that *GNPDA1* mRNA expression was significantly increased in hepatocellular cancer (HCC) tissues and hepatoma cell lines, and it is a hallmark of a poor prognosis [66,67]. Experimental findings obtained from shRNA-knockdown GNPDA1 revealed that the expression of this isoenzyme promoted cell proliferation and raised the percentage of cells in the G0/G1 phase, favoring cell migration and invasiveness. 

On the other hand, GNPDA1 knockdown in two different HCC cell lines increased the number of apoptotic cells, suggesting a possible role of GNPDA1 as an inhibitor of apoptosis [68]. Even though the mechanisms underlying these effects were not explored, the authors speculated that this increased GNPDA1 activity can provide an additional flux of Fru6P into the glycolytic pathway, favoring the energy supply in HCC cells. It is pertinent to emphasize that only the expression of GNPDA1, and not GNPDA2, was detected in the liver [5,26].

Furthermore, contradictory results were obtained when colorectal cancer SW480 cells were challenged with an anti-cancer activity compound derived from propolis [69]. As mentioned before, GNPDA1 displayed a pro-tumor effect in HCC, and we would expect that, by exposing cancer cells to an anti-tumor drug, the expression/activity of GNPDA1 would be down-regulated. Instead, this isoenzyme was up-regulated in SW480 cells treated with the anti-cancer drug. Notably, in addition to GNPDA1, the glycolytic enzyme glyceraldehyde-3-phosphate dehydrogenase (GAPDH) was also found to be up-regulated, reinforcing the proposed role of GNPDA1 as a “fuel supplier” for glycolysis. Nevertheless, for this type of cancer, this effect does not seem to be favorable. GAPDH has been implicated in some non-metabolic processes, for example, as an initiator of apoptosis [70]. This study did not suggest a reason for this; the fact is that colon and colorectal cancer SW480 cells express both GNPDAs, mostly GNPDA1, and its increased expression seems to be protective against tumor progression [69]. Therefore, more experimental studies are necessary to clarify this discrepancy. We can discern a possible explanation based on the metabolic roles of GFAT and GNPDA in UDP-GlcNAc6P and hyaluronan synthesis in human keratinocytes [2]. The following four enzymes catalyze the conversion of Fru6P into GlcN6P: GFAT1 and 2 and GNPDA1 and 2. When the biosynthesis of GlcNAc6P was blocked by silencing the *GFAT1* gene, the expressions of both GNPDAs increased; then, the synthesis of GlcN6P was catalyzed, which was likely the reason for stable levels of UDP-GlcNAc being maintained.

On the other hand, the depletion of both GNPDA1 and GNPDA2 resulted in a significant increase in the cell surface hyaluronan coating. Hyaluronan, or hyaluronic acid (HA), plays an essential role in the epithelial–mesenchymal transition (EMT), cell migration, and cancer metastasis [71]. As was mentioned previously, the silencing of *GFAT1* stimulated *GNPDA1* and *GDPDA2* expression, reducing cell migration [2]. Blocking HA synthesis in HCC Hepa 129 and Hep3B resulted in the inhibition of proliferation and increased apoptosis [72]. Whether a higher expression of GNPDA1 in hepatoma cells can increase HA synthesis, favoring pro-tumor effects, is still an open question. Finally, why the same protein behaves as a tumor suppressor or oncoprotein in different types of malignancies, as well as the mechanisms involved in this, are not entirely understood [73]. The existence of tissue-specific factors associated with proteins related to carcinogenesis and cancer development is a wide field to be explored.

## 6. A New Perspective on GNPDAs, O-GlcNAcylation, and Ammonium

A good start for this section is the sentence “Glucosamine-6-phosphate isomerase catalyzes the conversion of glucosamine 6-phosphate to fructose 6-phosphate”, as referred to elsewhere. Of course, this is essential from a catabolic point of view because of the central role of GNPDAs in the degradation of GlcN6P, determining the catabolic fate of UDP-GlcNAc. However, most studies forget NH_4_^+^, the other product of the deaminating aspect of the GNPDA reaction. In order to discuss the relevance of NH_4_^+^, first, we take a brief look at GlcNAc6P metabolism and the *O*-GlcNAcylation of proteins. UDP-GlcNAc is synthesized through the HBP (Figure 1). GlcNAc is a common metabolite shared by the synthesis of the *N*- and *O*-glycan pathways and the *O*-GlcNAcylation of proteins. *O*-GlcNAcylation serves as a glucose-sensing mechanism. When cellular glucose levels are elevated, glucose is diverted into the HBP, increasing the levels of UDP-GlcNAc and the *O*-GlcNAcylation of proteins [74]. There is growing evidence supporting the role of the *O*-GlcNAcylation of proteins in different cellular processes and cancer [1,75,76,77,78]. The *O*-GlcNAcylation of cytoplasmic, nuclear, and mitochondrial proteins is the dynamic and reversible addition of a molecule of β-D-*N*-acetylglucosamine to specific serine and threonine residues. Once UDP-GlcNAc is synthesized, a GlcNAc moiety is added to a protein by the enzyme *O*-linked *N*-acetyl-glucosaminyltransferase (OGT), while *O*-linked *N*-acetyl β-D-glucosaminidase (OGA) controls its removal [1]. Released GlcNAc can be phosphorylated and recycled back into UDP-GlcNAc, creating a salvage loop. Both OGT and OGA are dysregulated in cancer. In general, OGT is up-regulated and OGA is down-regulated [79]. In most cancers, *O*-GlcNAcylation is enhanced, yielding more *O*-GlcNAcylation of regulatory proteins. Recently, it was suggested that OGA is down-regulated both transcriptionally and by epigenetic mechanisms [80]. In this sense, higher *O*-GlcNAc levels induce the transcription of several proliferative genes through different signaling routes [81]; e.g., in breast cancer cells, the Ras–Raf–MEK–ERK signaling pathway is affected, because *O*-GlcNAcylation avoids the degradation of MEK2, which extends its proliferative effects [82]. *O*-GlcNAcylation contributes to therapeutic resistance in cancer cells through different mechanisms, ranging from those affecting the amount or quality of anti-cancer drugs to cellular processes like autophagy, apoptosis, genome stability, the epithelial–mesenchymal transition, and the cell cycle [79]. 

### Importance of GNPDA in Ammonium Metabolism

Traditionally, ammonia (NH_3_) and its protonated form (NH_4_^+^) were considered waste and toxic byproducts of amino acid and nucleotide catabolism, with NH_4_^+^ from GlnNAc6P catabolism also being identified recently. NH_4_^+^ toxicity is neutralized by integrating one molecule into glutamic acid (Glu), yielding Gln (the reaction is catalyzed by glutamine synthase; Figure 4), or even by the synthesis of urea in the liver. Then, urea is removed from the blood by the kidneys. Recently, the perception of NH_4_^+^ as a waste product of cellular metabolism has been changing, mainly with respect to cancer cells. Some mechanisms of ammonia metabolism supporting the growth and survival of cancer cells include (a) increased autophagy [83], (b) pyrimidine synthesis [84], (c) proline and aspartate synthesis [85], and (d) increased lipogenesis [86] (Figure 4). Indeed, it was found that NH_4_^+^ plays a key role as an activator of sterol regulatory element-binding transcription factor 1 (SREBP-1), promoting lipogenesis and tumor progression in different human cancer cells [86]. SREBPs are transcription factors that regulate the expressions of genes involved in cholesterol and lipid synthesis [87]. When cholesterol is high, SREBP-1 is located at the endoplasmic reticulum (ER) in a complex with SREBP-cleavage-activating protein (SCAP) and insulin-inducible gene protein (Insig) [88]. Experimental and computational data support that NH_4_^+^ released form Gln can bind SCAP through D428, releasing Insig and inducing the translocation of SREBP-1 from ER to the Golgi apparatus. In this organelle, SREBP1 is cleaved and the N-terminal domain is translocated into the nucleus, activating genes involved in lipid metabolism [86]. Increased lipogenesis is an essential hallmark of cancer progression and metastasis [89]. In this context, the enzymes involved in amino group metabolism have been intensely studied in neoplastic cells [90,91]. Due to the increased catabolism of Gln and decreased clearance caused by the poor vascularity inside solid tumors, cancer cells have higher levels of NH_4_^+^ compared with normal cells. The survival of cancer cells in high-NH_4_^+^ conditions depends on their ability to handle this excess by delivering NH_4_^+^ into anabolic pathways, providing building blocks such as nucleotides, amino acids, hexosamines, and indirectly, lipids, which are all necessary for an elevated rate of cell division (Figure 4).

As mentioned, under physiologic conditions, GlcN6P is synthetized from Fru6P and Gln by GFAT, and GNPDAs conduct the reversible deamination of GlcN6P. However, under some circumstances or metabolic conditions, GNPDAs can work in the anabolic direction, producing GlcN6P [13,14,64,65]. In this case, GNPDAs can incorporate NH_4_^+^ into the HBP. Exploring GNPDA’s role further, it is necessary to recall that the reaction catalyzed by GNPDAs is easily reversible (equilibrium constant of 0.2 M), while, on the other hand, the low *k_M_* for NH_4_^+^, as found for bGNPDA1 (3.7 mM), can push forward the biosynthetic function of GNPDA1 [18,30]. However, as mentioned previously, in ascites cancer cells, this enzyme functions mainly in the catabolism of GlcNAc, releasing NH_4_^+^ and incorporating the GlcN6P carbons into glycolysis and glycogen [64]. As represented in Figure 4, in cancer cells, the deaminating activity of GNPDA1 contributes to increasing the NH_4_^+^ cellular pool. At the same time, the enzyme supplies Fru6P to the glycolytic pathway, enhancing energy production [63] and fatty acid metabolism [91,92]. The link between GNPDA’s deaminating activity and the ammonium metabolism in cancer cells is still intriguing; a possible route for NH_4_^+^ coming from GlcN6P catabolism is its incorporation as the amino group for Glu or Gln by the reverse reaction of glutamate dehydrogenase [85,90,93,94] or by Gln-synthase [95,96]. In this way, the fate of NH_4_^+^ from GlcN6P could be in proteins, nucleic acids, or glutathione, which are necessary for cancer cell survival.

In conclusion, the role of GNPDA1 in cancer cells is currently unclear. The role of this enzyme in the synthesis or catabolism of GlcNAc clearly points toward an important function, in addition to GFAT1, in modulating the availability of GlcNAc for *O*-GlcNAcylation [7]. Dysregulation of the *O*-GlcNAcylation of proteins affects several cellular processes favoring cancer progression and other diseases, and both GNPDAs have kinetic properties that can support carcinogenic or anti-tumor mechanisms.

Despite the limitations of our current knowledge, we can still suggest some intriguing questions. Even when both deaminases catalyze the same reaction, why do GNPDA1 and 2 participate differentially in different diseases? Could these isoenzymes have other functions beyond their metabolic role? Moreover, is there any kind of epigenetic regulation operating on the expression of their genes? The first study aiming to explore the epigenetic regulation of the GNPDA2 gene was reported recently [97], and hopefully, more studies in this field will soon be produced. More experimental studies focused on these issues are necessary to clarify the roles of GNPDAs in metabolic and neoplastic diseases. 

## 7. Future Directions

Future research in this field, mainly on GNPDA2, must go further than epidemiologic and genomic association studies and explore the mechanisms involved in the changes in sugar metabolism implied in the pathogenesis of obesity and related diseases such as diabetes mellitus type 2, and also in the development of complications derived from the glycation of blood and endothelial proteins, considering the central role of Fru6P in this process. 

GNPDA2 is active in the hypothalamus, and its role in the mechanism of the regulation of glucose homeostasis is an open research field [61]. GNPDA1 is expressed in the brain to an even greater extent than GNPDA2; however, it is still unknown whether it plays any role in the metabolism of glucose in this organ in relation to obesity. In the same way, obesity is related to some types of cancer, such as breast cancer, and any possible involvement of GNPDA2 beyond GNPDA1 is yet to be identified

Another intriguing trait of GNPDA2 to be explored in the future is the existence of three isoforms. The human GNPDA2 variants show different numbers of amino acids, as follows: 276, 242, and 206 residues, corresponding to isoforms 1, 2, and 3, respectively. So far, there is no evidence regarding the different physiologic roles of these variants, or why GNPDA1 lacks isoforms.

The existence of natural hetero-hexameric forms of GNPDA (mixed oligomers of GNPDA1 and 2) has not been demonstrated. This indicates that evolutionary selection pressure has not led to monomer mixtures for either deaminase, and the reason for this is still unknown.

## Figures and Tables

**Figure 2 ijms-25-12054-f002:**
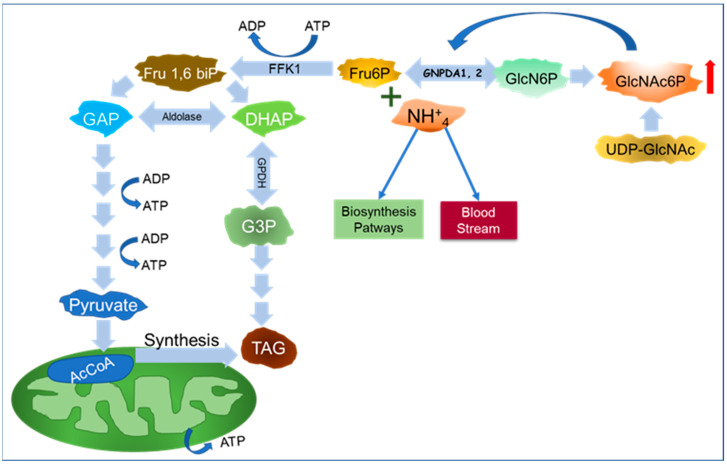
Graphic representation of GNPDAs’ role in producing energy and lipid synthesis. GNPDAs provide Fru6P for glycolytic pathways and ATP production. Fru6P is phosphorylated by Phosphofructokinase 1 (FFK1), yielding Fru1,6 biphosphate (Fru1, 6 bi P). Aldolase produces glyceraldehyde three phosphate (GAP) and dihydroxyacetone phosphate (DHAP). The glycolytic pathway generates ATP and pyruvate, which can enter the tricarboxylic cycle. Acetyl-coenzyme A (AcCoA) is the precursor of fatty acid synthase. The glycerol required for triacylglycerols is obtained from oxidation of DHAP into glycerol 3-phosphate (G3P). Increased GlcNAc6P activates GNPDAs.

**Figure 3 ijms-25-12054-f003:**
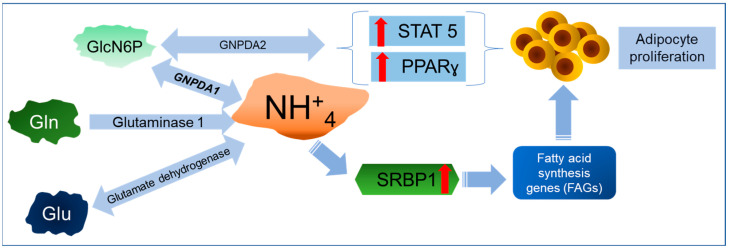
Graphic description of GNPDAs in adipocyte proliferation. GNPDA2, a key player in the metabolic process, significantly increases the levels of STAT5 and PPAR-γ, thereby stimulating the proliferation of adipose tissue cells. This pivotal role of GNPDA2 sheds light on the association of GNPDA2 with higher body mass index and obesity. The activation of both deaminases also contributes to the activation of fatty acid and cholesterol synthesis by increasing the amount of NH_4_^+^ and SRBP1 activation.

**Figure 4 ijms-25-12054-f004:**
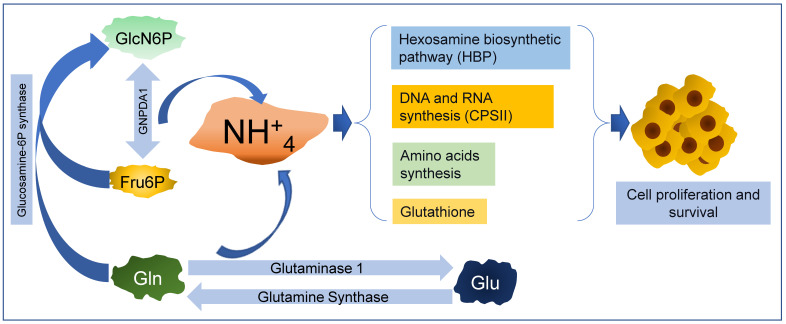
Graphic illustration of ammonium metabolism in a tumor cell, emphasizing the role of GNPDA1.Tumor cells are known for surviving in reduced blood supply conditions and hypoxic environments. The concentration of NH_4_^+^ can reach significantly high levels of 0.14–5 mM, compared to the 0.027–0.05 mM found in normal tissues [90]. The GNPDA1 catabolic reaction releases NH_4_^+^, as well as glutaminase 1, and contributes to increasing the intracellular concentration of NH_4_^+^. On the contrary, the fixing NH_4_^+^ reactions include (a) carbamoyl phosphate synthase II (CPS-II) and (b) GlcN6P synthase (GFAT). CPS-II and GFAT use Gln to synthesize carbamoyl-phosphate (CP) and GlcN6P, respectively. CP is used in the pyrimidine nucleotides biosynthetic pathway, yielding uridine triphosphate (UTP). Finally, UTP is utilized in the synthesis of UDP-GlcNAc. In addition, amino acids, glutathione, DNA, and RNA purine nucleotides can also be synthesized from Gln. They are necessary for cell proliferation and survival. These mechanisms decrease the potential toxic effects of NH_4_^+^.

**Table 1 ijms-25-12054-t001:** Summary of diseases associated with GNPDA isoenzymes.

Enzyme	Related Diseases	Kind of Study	References
GNPDA2	Obesity and asthma	Epidemiologic ^a^; no experimental data	[37,38]
GNPDA2	Obesity and type 2 diabetes	Epidemiologic and experimental ^b^	[39,40,41,42,43,44,45,46,47,48,49,50,51,52,53,54,55,56,57,58,59]
GNPDA2	Obesity and neurologic diseases	Meta-analysis	[59,60]
GNPDA2	Metabolic control and neurologic diseases	Experimental ^c^	[61,62,63]
GNPDA1	Gastrointestinal cancer, hepatocellular cancer, and melanoma	Experimental ^d^	[64,65,66,67,68,69]

GNPDA2 is related to obesity, while GNPDA1 expression is altered in hepatocellular carcinoma and colon cancer. The lack of experimental studies clarifying the role of GNPDA2 in obesity is significant. ^a^ Epidemiologic studies include GWASs, genotyping SNP associations, observational, and genetic associations. ^b^ The only experimental study [52]. ^c^ Silencing and over-expression of GNPDA2, proteomic analyses, and transgenic mouse model. ^d^ In vitro metabolic studies with cell lines and tissue extracts, silencing, over-expression of genes, and pharmacologic treatments.

## Data Availability

Not applicable.

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
