# Peer review of "The Possible Roles of Glucosamine-6-Phosphate Deaminases in Ammonium Metabolism in Cancer"

_ijms, 2024, doi:10.3390/ijms252212054_

Round 1
Reviewer 1 Report
Comments and Suggestions for Authors
The article titled "The Possible Roles of Glucosamine-6-Phosphate Deaminases in Ammonium Metabolism in Cancer," presented by Lara-Lemus et al. emphasizes the significance of glucosamine-6-phosphate deaminases in ammonium metabolism in cancer cells, thereby contributing meaningfully to current cancer research. The investigation of ammonium metabolism as a metabolic by-product, together with its deeper ramifications beyond mere waste, is innovative and significant. The review is structured, presenting GNPDA's function in hexosamine production, its enzymatic activity, and its contribution to ammonium metabolism in cancer. I believe that the authors have carefully examined the distinctions between the two isoforms and their implications in various illnesses. Some areas that require modification before being accepted for publication.
1. Some sections, particularly those discussing the kinetic properties and allosteric regulation of GNPDA1 and GNPDA2, are overly technical. The terms "K-effect," "V-effect," and "antiergistic K-V+" are complex and may be unfamiliar to readers not specializing in enzyme kinetics. Please provide a more simplified explanations or include a brief glossary of terms that would enhance readability.
2. While the link between GNPDA activity and ammonium metabolism is intriguing, the revision may benefit from a more in-depth exploration of how higher ammonium levels alter cancer cell behavior. The study indicates that ammonium can activate sterol regulatory elements and increase lipogenesis, but more mechanistic insights into how this drives tumor progression, or why cancer cells are so good at handling ammonium, would be helpful.
3. The manuscript discusses GNPDA1's paradoxical activities in cancer, including its putative preventive effect in colorectal cancer and tumor-promoting role in hepatocellular carcinoma (HCC). While noting these conflicts is important, the review might go into greater detail about the possible causes of these differences. Hypotheses about context-specific activity or tissue-specific effects could be developed to guide future research efforts.
4. Table 1, which lists disorders related with GNPDA isoenzymes, is useful. A clearer demarcation between findings from genetic association studies (e.g., GWAS) and experimental data would be advantageous. The table could also indicate whether the relationships are purely correlative or if causal ties have been identified.
5. Perhaps a "Future Directions" section could strengthen the manuscript's conclusion and motivate additional research.
Author Response
Please see the attacment.

Reviewer 2 Report
Comments and Suggestions for Authors I highly value the authors' proactivity in preparing a review manuscript on the Glucosamine-6-Phosphate Deaminases in Ammonium Metabolism in Cancer. Without a doubt, I consider that the manuscript is likely to have a considerable impact, in addition to occupying a well-deserved place as a reference in other research works that deal with results related to any of the aspects reviewed. please improve the review based on the comments* The introduction is too brief and does not provide a comprehensive overview of the review. It is recommended to expand the introduction to cover the key aspects of the research. Additionally, labeling the introduction as section "1" would be more appropriate.
* Abbreviations should be spelled out in full ONLY upon their first appearance in the text, and then the abbreviated form should be used consistently throughout the rest of the paper, there are repeated abbreviations.
* The fonts in figure1 and 2 is not profession ARIAL or new roman times.Besides, the legend is too brief to act as a clear description of the figures.
* While the paper contains a significant amount of content, there are missing a figures to clearly illustrate and support the points being made in section 3 and 5. It is recommended to include more figures to present the information in a clearer and more straightforward manner.
* Copyright issues should be taken into account when creating figures. So It is advisable to mention the use of software tools in the generation of figures to ensure proper attribution and compliance with copyright requirements.
* In the abstract part , the authors should mention the unique points of this review compare to other reviews.
* LINE 162-171 is the summary of part2 and opinions of authors? If not, there should be some references.
* 5.1 is suggested to incorporate with the paragraph which close to 5. 5. A new perspective on GNPDAs, O-GlcNAcylation, and ammonium. And then it is reasonable to have a figure to represent the entire part 5 Comments on the Quality of English Language
good
